# Improved Fluorescence and Gain Characteristics of Er-Doped Optical Fiber with PbS Nanomaterials Co-Doping

**DOI:** 10.3390/ma15176090

**Published:** 2022-09-02

**Authors:** Xiangping Pan, Yanhua Dong, Jianxiang Wen, Yana Shang, Xiaobei Zhang, Yi Huang, Fufei Pang, Tingyun Wang

**Affiliations:** Key Laboratory of Specialty Fiber Optics and Optical Access Networks, Joint International Research Laboratory of Specialty Fiber Optics and Advanced Communication, Shanghai Institute for Advanced Communication and Data Science, Shanghai University, Shanghai 200444, China

**Keywords:** PbS nanomaterials, PbS/Er co-doped optical fiber, energy level splitting, fluorescence characteristics

## Abstract

Er-doped optical fiber (EDF) with ultra-broad gain bandwidth is urgently needed given the rapid advancement of optical communication. However, the weak crystal field of the host silica glass severely restricts the bandwidth of traditional EDF at 1.5 μm. In this study, we theoretically explored the introduction of PbS nanomaterials in the silica network assisted with the non-bridging oxygen. This can significantly increase the crystal field strength of Er^3+^ ions in the local structure, leading to their energy level splitting and expanding the fluorescence bandwidth. Additionally, the PbS/Er co-doped optical fiber (PEDF) with improved fluorescence and gain characteristics was fabricated using modified chemical vapor deposition combined with the atomic layer deposition technique. The presence of PbS nanomaterials in the fiber core region, which had an average size of 4 nm, causes the ^4^I_13/2_ energy level of Er^3+^ ions to divide, increasing the fluorescence bandwidth from 32 to 39 nm. Notably, the gain bandwidth of PEDF greater than 20 dB increased by approximately 12 nm compared to that of EDF. The obtained PEDF would play an important role in the optical fiber amplifier and laser applications.

## 1. Introduction

Er-doped silica fiber (EDF) was firstly prepared in 1985, which considerably promotes the development of rare-earth-ions-doped active optical fiber [1]. The 1.5-micrometer fluorescence emission of Er^3+^ ions have been rapidly developed and applied in fiber lasers and fiber amplifiers because it coincides well with the lowest optical communication window of silica fiber [2,3,4,5]. Recently, the amplification performance of EDF in the C-band of optical communication has been unable to meet the increasing demand for high-speed communication in the global communication system. The gain bandwidth (GB), noise figure (NF), and power conversion efficiency (PCE) of Er-doped fiber amplifiers (EDFA) have thus been continuously improved by researchers through rare-earth ion co-doping [6,7], matrix material optimization [8,9,10], and structural design [11,12]. Although it can be improved by the non-uniform broadening effect of high aluminum (Al) dopant [13], the GB of EDF is strongly limited by the relatively weak crystal field of host glass materials coupled with the low PCE.

In the near-infrared region, nano-semiconductor materials (e.g., PbS) have excellent fluorescence and structural properties, such as wide bandwidth, tunable band gap, and highly symmetric structure [14,15]. Moreover, many researchers have tried to introduce nano-semiconductor materials into rare-earth-doped fibers to boost the GB of the fiber by increasing the fluorescence bandwidth and local crystal field. Zhang et al. explored the local crystal field effect of semiconductor nanomaterials on the fluorescence properties of Er^3+^ ions at 1.5 μm and the near-infrared fluorescence properties of PbS and Er co-doped silica glass materials in 2015 [16,17]. Additionally, Wan et al. reported the co-doped glasses with PbS nanocrystals and Tm^3+^ ions. After that, they changed the size of nanocrystals to increase the fluorescent efficiency and bandwidth of Tm^3+^ ions in the near-infrared region [18]. Moreover, PbS nanocrystals were used as the saturable absorber to broaden the bandwidth of Tm-doped fiber lasers, which is important for the passively mode-locking lasers in mid-infrared bands [19]. However, the interaction mechanism between nano-semiconductors and rare-earth ions in the local structure of silica glass remains to be studied.

Benefiting from the rapid development of density functional theory (DFT), it provides a theoretical approach for comprehending the atomic-level interaction between nano-semiconductors and rare-earth ions [20,21]. Additionally, nano-semiconductors and rare-earth ion co-doped silica fibers have been rarely reported due to the limitations in fiber preparation techniques. Atomic layer deposition (ALD) has significant benefits in the preparation of nano-semiconductors and rare-earth ions co-doped silica fibers owing to its high uniformity and conformity in the deposited nanofilms [22,23].

In this paper, the local structural models of PbS-doped and PbS/Er co-doped silica fiber materials were established for the first time. Based on DFT, the interaction between PbS and the silica network, and the local crystal field effect of PbS on the fluorescence properties of Er^3+^ ions in co-doped local structures are theoretically studied. Subsequently, the PbS/Er co-doped silica fiber (PEDF) was obtained by the ALD combined with the modified chemical vapor deposition (MCVD) technology, and the effect of the PbS nanomaterials on the amplification performance of EDF was investigated in terms of microstructure, fluorescence properties, gain, and NF.

## 2. Theoretical Calculations

### 2.1. Local Structural Models of PbS-Doped Fiber Materials

Silica glass has an amorphous network structure with the characteristics of the long-range disorder and short-range order. Three-membered rings (3MRs), four-membered rings (4MRs), five-membered rings (5MRs), six-membered rings (6MRs), and other hybrid ring units make up the majority of the silica glass microstructure [24]. It is important to optimize the computational complexity for large-size atomic–molecular structures. As the simplest and most common network structure, 3MRs make up a significant component of the network structure and are frequently employed to define the structure and optical properties of silica-doped materials in conjunction with economic calculation [25,26,27,28]. Therefore, the 3MR structure is used as the fundamental silica glass network to examine the interaction between PbS nano-semiconductors and silica network, and its impact on the local structure and fluorescence properties of doped Er^3+^ ions by DFT. Although the atomic–molecular structure has a relatively small number of atoms, it can approximately show the band gap distribution range of PbS nano-semiconductor materials [29,30]. Finally, the energy level structure of PbS/Er co-doped silica fiber material was established.

Firstly, the ground-state local structural models were optimized using the Becke-3-Lee-Yang-Parr (B3LYP) hybrid function in the Gaussian-09 program [31]. The 6–31 + G** basis sets are used for H, O, Si, and S elements, and Pb and Er are replaced by the 4 and 11 valence electrons relativistic effective core potentials (RECPs), respectively [32,33]. Following optimization, the excited state characteristics of the local structures are analyzed using the time-dependent density functional theory (TD-DFT) [34].

The connection of PbS with the silica network would break or recombine the silica lattice structure, thereby forming much non-bridging oxygen (NBO) in the host glass. The three potential local structural models for the incorporation of PbS into the silica glass were established based on the 3MRs, as shown in Figure 1. In model (a), PbS as a modifier is embedded into the 3MR. In model (b), the Si tetrahedron and PbS combine to create a new ring out of the 3MR. According to model (c), PbS directly connects with the Si tetrahedron by the bridging oxygen (BO), accompanied by the generation of NBO. The ground-state level energy of the three structural models were determined through DFT optimization. Combined with the equation for calculating bonding energy (eV) [26]:(1)$$E=nE\left(Si\right)+mE\left(O\right)+kE\left(H\right)+E\left(Pb\right)+E\left(S\right)-E\left(Si_{n}O_{m}H_{k}PbS\right)$$
where n, m, and k are the number of *Si*, *O*, and *H* atoms in the models, respectively. As indicated in Table 1, the calculated bond energies for models (a), (b), and (c) are 7.3141, 6.0843, and 7.3265 eV, respectively. Since the bonding energies of the obtained structures are relatively small, it can be explained by the incorporation of PbS into the silica network, which makes the local structures unstable. Model (c) exhibits the highest bonding energy, indicating that it is reasonably stable.

To further demonstrate the validity of PbS-doped 3MR, the absorption and fluorescence characteristics are analyzed using TD-DFT. Based on the calculations of the ground and excited states, the absorption shoulder of PbS-doped 3MR appears at 747 nm, which is caused by the excitonic absorption of PbS nanocrystals, as shown in Figure 2a. Moreover, the calculated fluorescence peak of the PbS-doped 3MR structure appears at 1036 nm (Figure 2b), which is in good agreement with the previous experimental results [35]. Therefore, the local structure model (c) is relatively reliable and valid to study the effect of PbS co-doping on the local structure and energy level of Er^3+^ ions.

### 2.2. Local Structural Models of PbS/Er Co-Doped Fiber Materials

According to the previous report [36], it is difficult for Er^3+^ ions to directly embed into the silica network due to its large ionic radius. Therefore, it will break the original network structure and connect with the Si tetrahedron of 3MRs through Bos. To balance the local electronic valence, other Si tetrahedron are also coupled to the Er^3+^ ion. However, the length of BO bonds between the Si tetrahedron and Er^3+^ ion becomes longer owing to the incorporation of PbS in the optimal structure. Furthermore, Er^3+^ ions primarily function as network modifiers to engage in interactions with the silica matrix via NBOs as opposed to Bos [37]. In order to change the valence balance, the local structure surrounding the Er^3+^ ion may be damaged through the creation of NBOs. Therefore, based on model (a), another optimized structural model (b) of PEDF material is also established, as shown in Figure 3. The two Si tetrahedrons outside the 3MR connected to Er^3+^ ions are removed. Moreover, new NBOs are added to enhance the electronegativity around Er^3+^ ions and balance the influence of doping PbS nanomaterials.

The excited state parameters of models (a) and (b) for PEDF are calculated using TD-DFT. Table 2 and Table 3 provide a list of their electronic transitions, oscillator strengths, and configurations. For model (a), there are mainly excitation energy levels of 0.7878, 1.2934, 1.3664, 1.5441, and 1.8694 eV, and the corresponding oscillator strengths (f) are 0.0002, 0.0009, 0.0005, 0.0017, and 0.0009, respectively. It is discovered that the direct addition of PbS material on the Er-doped 3MR structure does not change the excitation state characteristics of Er^3+^ ions in the 1.5-micrometer band, and the corresponding oscillator intensity near 1573 nm is weak at 0.0002.

For model (b), the excitation energy level (^4^I_13/2_) of Er^3+^ ions at 1.5 μm changes from one level (1573 nm) to two levels (1533 and 1595 nm). The obvious Stark level splitting of Er^3+^ ions is mainly caused by the presence of PbS semiconductor materials, which not only boosts electron charge, but also enhances the local crystal field around Er^3+^ ions. Moreover, the oscillator strength of the excited state at 1533 and 1595 nm is significantly increased, reaching 0.0043. The obtained results indicate that the introduction of PbS in silica glass materials can alter the local coordination environment of Er^3+^ ions. As a result, this leads to a broadening of the spectrum bandwidth and an increase in the intensity of the oscillator of the excitation energy level, which optimizes the fluorescence efficiency of the Er^3+^ ion.

### 2.3. Frontier Molecular Orbital

To further illustrate the regulation effect of the PbS materials on the Er^3+^ ion-doped silica material, the energy band structure and electron cloud density distribution for models (a) and (b) were compared and analyzed, as shown in Figure 4. Frontier molecular orbitals are defined as the highest occupied molecular orbital (HOMO) and lowest unoccupied molecular orbital (LOMO), where the band gap is the difference in energy between the two. For model (a), the frontier molecular orbital energy values from HOMO^−3^ to LOMO^+3^ are −17.207, −17.025, −16.933, −16.349, −13.853, −13.721, −13.302, and −13.178 eV, respectively. For model (b), the frontier molecular orbital energy values from HOMO^−3^ to LOMO^+3^ are −18.608, −18.242, −18.149, −18.100, −15.156, −14.753, −14.658, and −14.461 eV, respectively. The band gaps corresponding to models (a) and (b) are 2.5 and 2.9 eV, respectively, owing to the doping of Er^3+^ ions and PbS materials [26]. Furthermore, regardless of the HOMO or LOMO, the electron cloud distribution density around Er^3+^ ions in model (b) is significantly higher than that in model (a). The results further demonstrate that the introduction of PbS nanomaterials combined with NBO can not only provide a better valence-balanced environment, but also allow the local structure’s charge to accumulate around the Er^3+^ ions. As a result, Er^3+^ ions have increased activity and take on the role of the active center for the entire local structure. Importantly, in the LOMO state of model (b), the Er^3+^ ions are completely covered by the electron cloud after the introduction of PbS and NBO, which increases the strength of the local crystal field and the probability of Stark level splitting.

### 2.4. Energy Level

According to the calculated excited state level and the corresponding oscillator strength, the energy level structure of the PEDF material can also be established, as shown in Figure 5. The black arrow line in the figure represents the absorption process of Er^3+^ ions. The red solid line represents the splitting energy level of the Er^3+^ ion ^4^I_13/2_ level due to the co-doping of PbS materials. The dashed line with the purple arrow shows the excited state transition process. The solid blue arrow line shows the transition luminescence process of Er^3+^ ions. It can be shown that the possible emission wavelength range of the PEDF material is approximately 1533–1595 nm, and its spectral bandwidth can cover 62 nm, which is significantly larger than 38 nm in the EDF material. It provides a theoretical foundation for the development of ultra-broad and high-performance EDFA.

## 3. Experimental Section and Results

### 3.1. Fiber Preparation

In this work, PEDF and EDF samples were fabricated ALD technique combined with the MCVD method, as shown in Figure 6. Firstly, HF solution was processed with to remove impurities from the pure silica tube, which had an outer diameter of 30 mm, an interior diameter of 25 mm, and a length of 21 cm. Secondly, the high-purity precursor materials (SiCl_4_) volatilized and was carried into the rotating silica tube by a carrier gas (O_2_). The gas mixture was oxidized in the hot zone of a hydrogen/oxygen (H_2_/O_2_) burner from the outside at a temperature of approximately 1800 °C. The doped SiO_2_ particles were deposited on the inner wall of the silica tube in the form of fine soot, which eventually formed porous layers. During the subsequent doping process, the porous soot layers can effectively avoid RE ion clustering behavior and optimize the uniformity of the doped elements [35]. Thirdly, the ALD technique (Beneq TFS-200, Finland) was used to alternately deposit PbS and Er_2_O_3_ nanofilms on the surface of the porous layer. For PbS nanomaterials, bis(2,2,6,6-tetramethyl-3,5-heptanedionato)lead(II) (Pb(tmhd)_2_) and H_2_S gas were used as Pb and S precursors, respectively. The deposition process of PbS has been described in the literature [32]. For Er_2_O_3_, tris(2,2,6,6-tetramethyl-3,5-heptanedionato)erbium(III) (Er(tmhd)_3_) and O_3_ gas were used as Er and O precursors, respectively. The evaporation temperature range of Er(tmhd)_3_ was 145–170 °C, and the reaction temperature range was 250–350 °C. Then, Er(tmhd)_3_ (0.8 s duration) and O_3_ (1 s duration) pulses were alternated to produce Er_2_O_3_ nanofilms. The purging time between each precursor pulse was 2 s. After that, as a protective layer, Al_2_O_3_ was deposited by ALD on the surface of the PbS and Er_2_O_3_ nanofilms, preventing the volatilization of PbS and Er_2_O_3_ during the high-temperature MCVD process [38,39]. The precursor sources of the Al_2_O_3_ process are Al(CH_3_)_3_ and O_3_, respectively. The deposition parameters have been described in the literature [35]. Fourthly, co-doped materials such as GeO_2_ and SiO_2_ were deposited as the core layer using MCVD technology. The soot was subsequently consolidated into a clear glass layer using the moving burner. The tube was collapsed at very high temperatures of around 2250 °C to a transparent fiber preform with a diameter of approximately 16 mm. Finally, the fiber preform was put into the drawing tower to make the PEDF samples with core and cladding diameters of approximately 8.85 and 125.25 μm, respectively, as shown in Figure 6h. To further demonstrate the incorporation effect of PbS nanomaterials on the fluorescence characteristics of EDF, we prepared an EDF sample without PbS nanomaterials using the same preparation process. The diameter of the core and cladding of EDF were 8.66 and 124.93 μm, respectively, as shown in Figure 6i.

### 3.2. Structural Properties

X-ray energy spectrometer (EDS, MX80, Oxford Instruments, Oxford, UK) was used to determine the specific contents of different elements in the core region of the optical fiber. The results are listed in Table 4. The weight percent ratios (wt.%) of Pb and Er in PEDF are 0.98% and 0.12%, respectively. Additionally, there was essentially no difference in the content of Er and Al elements in the two samples.

The relative refractive index difference (RID) of the fiber samples was ascertained using the fiber refractive index analyzer (S14, Photon Kinetics, Beaverton, OR, USA), as depicted in Figure 7. The insets are cross-sectional views of the two samples. Moreover, the RIDs of EDF and PEDF are 1.08% and 1.17%, respectively. The results show that the RID of optical fiber can be effectively improved by adding PbS nanomaterials into the optical fiber core, which is also consistent with the previous conclusion [22].

To determine the distribution of PbS and Er materials, the focused ion beam (FIB) micro-cutting technology (600I, FEI, Brno, Czekh) was utilized to cut the thickness of less than 50 nm from the vertical surface of the fiber core region. Then, a standard sample with a cross-sectional area of about 10 × 2 × 3 μm was used to observe the core region of the PEDF with high-resolution transmission electron microscopy (HRTEM, JEM-2010F, JEOL, Tokyo, Japan), as shown in Figure 8a. The nanoparticles in the core region were uniformly distributed with an average size of approximately 4 nm, and the lattice fringe spacing was 0.28 nm. The diffraction rings measured in the selected area electron diffraction (SAED) diagram of Figure 8b correspond to the crystal planes (200) and (220) of PbS, which further confirms that the observed crystalline phase structure is attributed to PbS nanoparticles. Then, the element distribution of the fiber core was discovered using EDS analysis (Figure 8c). The distribution of Er elements in the PEDF was relatively uniform without obvious clustering and enrichment, which further proves the advantages of the MCVD/ALD method.

### 3.3. Fluorescence Properties

The cut-off method [22] was used to analyze the absorption characteristics of the samples, and the results are displayed in Figure 9a. Four typical absorption peaks associated with Er^3+^ ions were observed, located at approximately 650, 800, 980, and 1535 nm, respectively. These absorption peaks were mainly attributed to the level transitions of Er^3+^ ions at ^4^I_15/2_→^4^F_9/2_, ^4^I_15/2_→^4^I_9/2_, ^4^I_15/2_→^4^I_11/2_, and ^4^I_15/2_→^4^I_13/2_ [40]. For PEDF, two weak absorption peaks at approximately 753 and 1124 nm were also observed, which can be attributed to PbS nanomaterials [22]. Due to the small crystal size, the absorption of PbS nanocrystals mainly appeared in the ultraviolet to the visible light band, while it was weak in the near-infrared (NIR) region [41,42,43]. Therefore, although the weight ratio of Pb in the fiber was higher than that of Er^3+^, the absorption capacity of PbS nanomaterials in the NIR band was still significantly weaker than that of Er^3+^ ions.

The excitation and emission spectra of the fiber samples were examined using the Edinburgh FLS-980 fluorescence spectrometer to further characterize the fluorescence properties, as shown in Figure 9b. The EDF sample had an obvious excitation peak at 980 nm, and the corresponding emission peak was located at 1530 nm. The full width at half maximum (FWHM) of emission spectrum was approximately 32 nm, which is attributed to the ^4^I_13/2_→^4^I_15/2_ energy level transition. According to Figure 9c, the emission peak of PEDF near 1530 nm was broadened, and the FWHM was raised to approximately 39 nm. The emission peaks can be fitted with Gaussian functions to provide two fitted emission peaks at 1532 nm and 1541 nm. It indicates that the introduction of PbS nanomaterials can enhance the crystal field strength around Er^3+^ ions, resulting in the Stark splitting of ^4^I_13/2_→^4^I_15/2_ energy levels. This is consistent with the previous simulation results. Compared with EDF, the FWHM of PEDF increases by about 22%, indicating that the introduction of PbS nanomaterials is beneficial to EDF to achieve wider gain bandwidth. However, Er doped in areas where PbS does not exist (Er-3MR) will have its original fluorescence characteristics. With the fluorescence superposition of the Er^3+^ ions in the nanoparticle-free region, it leads to the obtained PEDF spectrum being broadened by 7 nm, which is less than the 24 nm in the theoretical calculation. In addition, the wavelength of the splitting peak at 1541 nm is smaller than the theoretically predicted 1595 nm. Therefore, it can be speculated that both PbS/Er-3MR and Er-3MR structures are present in the PEDF, which leads to deviations in the calculated and experimental results.

The fluorescence lifetime of EDF and PEDF samples are depicted in Figure 9d. Based on the excitation and emission spectra, the excitation and monitored emission wavelengths were chosen to be 980 and 1530 nm, respectively. The fluorescence lifetime of EDF and PEDF are 10.5 and 10.9 ms, respectively. With the introduction of PbS nanomaterials, the fluorescence lifetime of Er^3+^ ions increases. This indicates that the non-radiative transition process of Er^3+^ ions in PEDF samples is suppressed, which would increase excited electrons in the metastable ^4^I_13/2_ level. As a result, the probability of stimulated radiation is increased, which is more conducive to the improvement of optical fiber amplification performance.

### 3.4. Gain Properties

To confirm the impact of PbS nanomaterials on the fluorescence properties of Er^3+^ ions, a 980-nanometer pump source was utilized to monitor the fluorescence variations in the range of 1510–1580 nm, as shown in Figure 10a. The length of the fiber samples is 7.5 m. The fluorescence intensity of PEDF is higher than that of EDF at the same pump power, and the spectral width is also broadened. Figure 10b displays the fluorescence peak intensities of EDF and PEDF at 1.55 μm under different pump powers, further demonstrating the structural regulatory effect of PbS nanoparticles on the silica network that results in an increase in EDF fluorescence intensity.

The backward pump system is used to amplify the optical signal and analyze the gain performance of the fiber sample. The system diagram is shown in Figure 11a. The system consists of the 1480–1640-nanometer band tunable laser (TSL-710, SANTEC, Komaki, Japan), 1550-nanometer band optical fiber attenuator, 980-nanometer single-mode laser diode (LD), 980/1550-nanometer wavelength division multiplexer (WDM, THORLABS, Newton, NJ, USA), isolator at 1550-nanometer band (ISO, THORLABS, Newton, NJ, USA), and spectrum analyzer (YOGAWA AQ6370D, Tokyo, Japan).

The gain characteristics of the two fiber samples are displayed in Figure 11b. The pump power and injected signal light intensity are approximately 248.1 mW and −30 dBm, respectively. PEDF exhibits a greater and wider gain than EDF, with a maximum gain of about 46 dB at 1535 nm. Notably, the GB of PEDF greater than 20 dB increases by about 12 nm compared with EDF. The NF of the two fiber samples is also shown in Figure 11c at various pump powers. It can be seen that the NF of PEDF is slightly lower than that of EDF. Moreover, the lowest NF of PEDF is approximately 4.8 dB, which is suitable for the demand of optical amplifiers. Therefore, it can be speculated that the addition of PbS optimizes the local coordination environment of Er^3+^ ions, thereby reducing NF in EDFA.

## 4. Conclusions

In this study, the local structural models of PbS-3MR and PbS/Er-3MR were established, and their related structure parameters were calculated using Gaussian-09 by DFT and TD-DFT. We theoretically investigated that the energy level of Er^3+^ ions at the 1.5-micrometer fluorescence band was splitting into two levels, resulting from enhanced local crystal field strength in PbS/Er-3MR structural models assisted with the NBO. Furthermore, a novel PEDF embedded with PbS nanomaterials was prepared using the MCVD method combined with the ALD technique. The incorporated PbS nanomaterials were confirmed by TEM with an average size of approximately 4 nm. As a result, the fluorescent FWHM of PEDF at 1.5 μm was increased to 39 nm due to the Stark energy level splitting, showing an improvement of 22% over that of EDF. Meanwhile, the fluorescence lifetime of PEDF was also improved to 10.9 ms. The gain of PEDF at 1535 nm reached 46 dB, and the GB of PEDF greater than 20 dB increased by almost 12 nm compared to that of EDF. It is believed that the obtained PEDF coupled with the nano-semiconductors would be promising candidates for fiber amplifiers, lasers, and broadband light sources.

## Figures and Tables

**Figure 1 materials-15-06090-f001:**
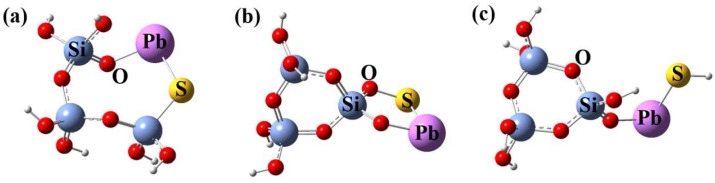
Local structural models of PbS-doped 3MR. (**a**) PbS embedded into the 3MR; (**b**) PbS combined with Si tetrahedron; (**c**) PbS connected with Si tetrahedron by BO.

**Figure 2 materials-15-06090-f002:**
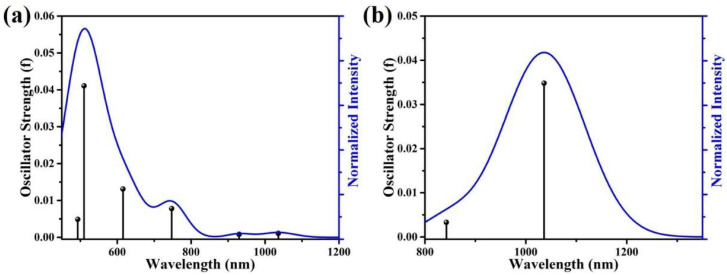
(**a**) Calculated absorption spectrum of PbS-doped 3MR; (**b**) calculated fluorescence spectrum of PbS-doped 3MR.

**Figure 3 materials-15-06090-f003:**
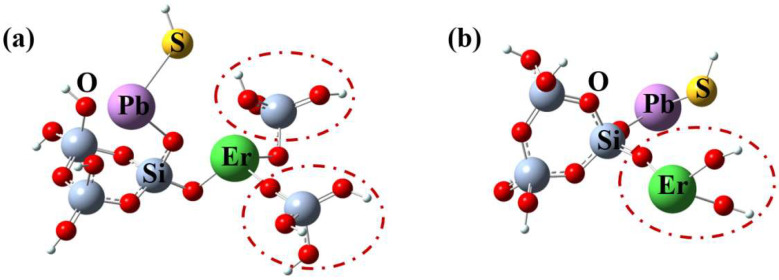
Local structural models of PbS/Er co-doped 3MR. (**a**) Er connected with two Si tetrahedrons; (**b**) Er connected with two BOs.

**Figure 4 materials-15-06090-f004:**
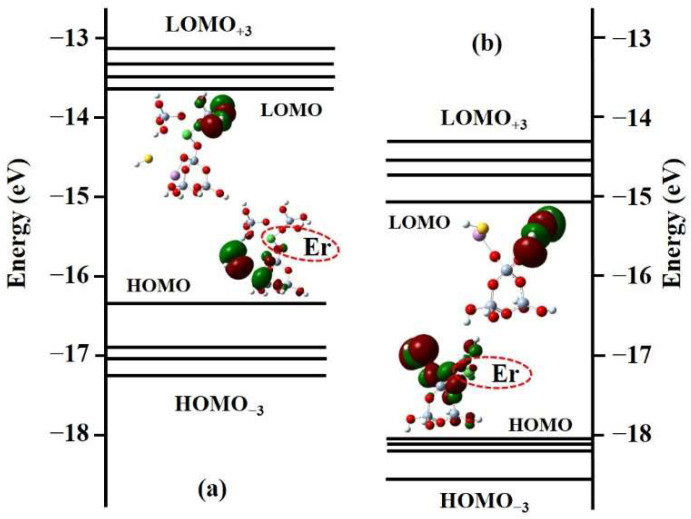
Energy eigenvalues and electron density distributions of the HOMO and LUMO; (**a**) Er-3MR structural model; (**b**) spin-up state of PbS/Er-3MR structural model.

**Figure 5 materials-15-06090-f005:**
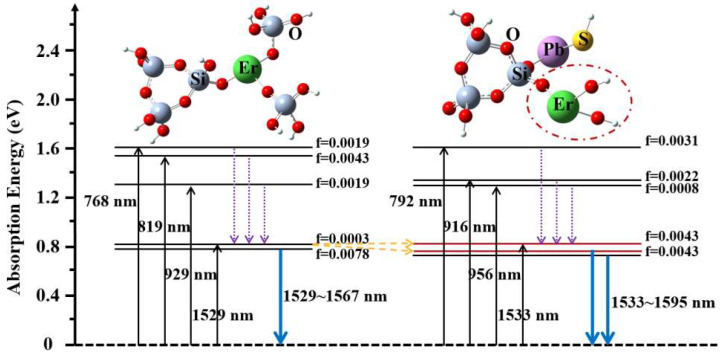
Energy level diagrams of Er-3MR and PbS/Er-3MR local structural models.

**Figure 6 materials-15-06090-f006:**
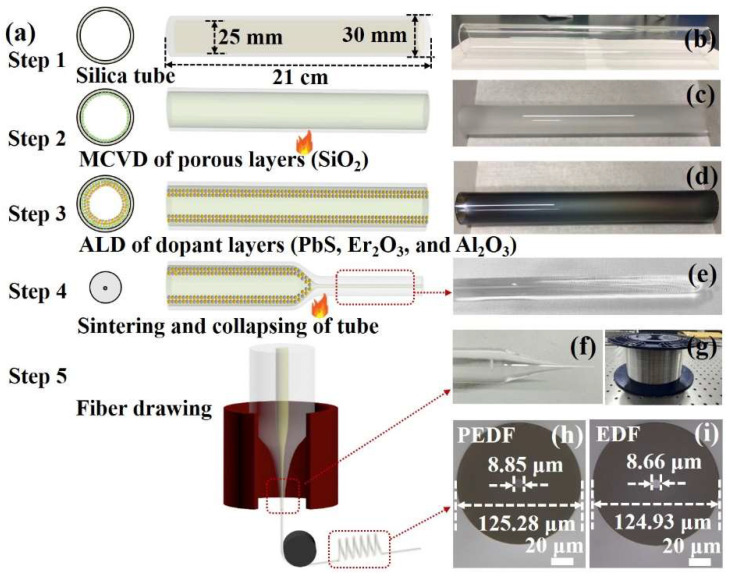
(**a**) Schematic diagram of fiber fabrication process; (**b**) silica tube after HF solution treatment; (**c**) silica tube after deposition of SiO_2_ porous soot layers; (**d**) silica tube after ALD process; (**e**) optical fiber preform; (**f**) fiber preform after drawing; (**g**) fiber samples; (**h**) the cross section of PEDF; (**i**) the cross section of EDF.

**Figure 9 materials-15-06090-f009:**
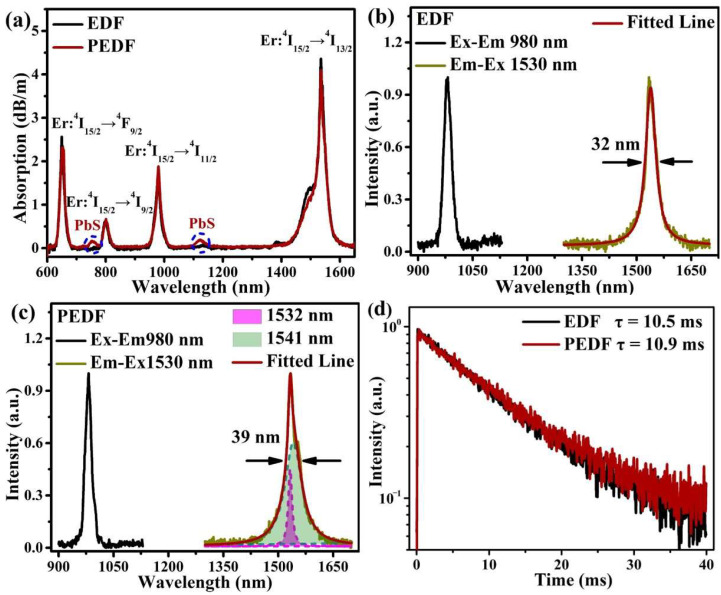
(**a**) Absorption spectra; (**b**) excitation and emission spectra of EDF; (**c**) excitation and emission spectra of PEDF, and the mauve and cyan-blue dot lines corresponding to the base lines of the fitted peaks at 1532 and 1541 nm, respectively; (**d**) fluorescent lifetime curves.

**Figure 10 materials-15-06090-f010:**
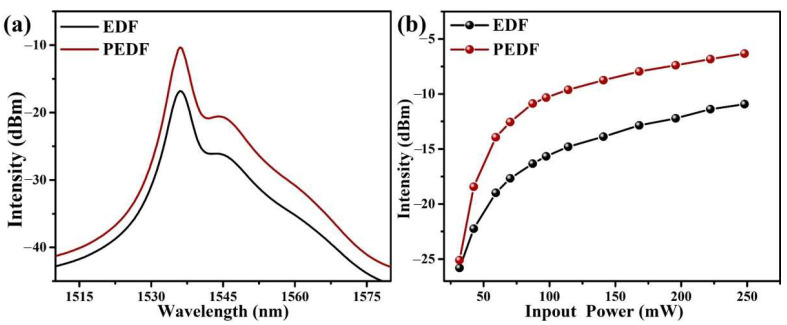
(**a**) The fluorescent spectra of EDF and PEDF under the pump at 980 nm with the length of 7.5 m; (**b**) the relationships between the input power and the luminescence intensities for EDF and PEDF.

**Figure 11 materials-15-06090-f011:**
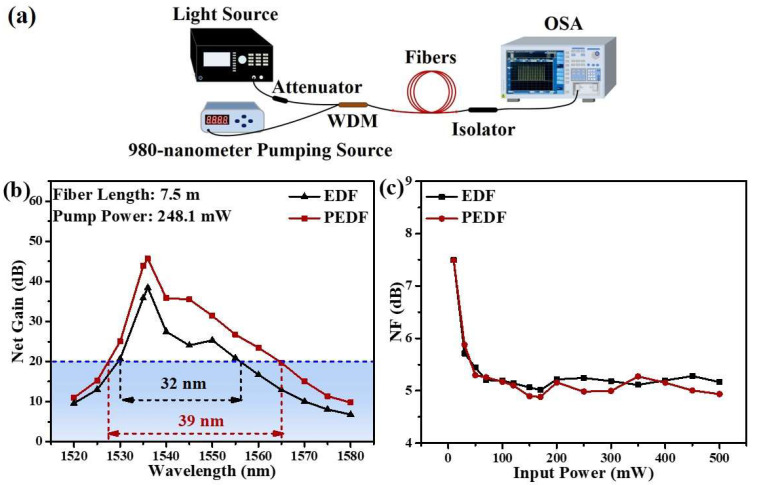
(**a**) Schematics of the 1.5-micrometer band fiber amplifier; (**b**) gain characteristics for two fibers; (**c**) the variation of NF with the different input powers for EDF and PEDF.

**Figure 7 materials-15-06090-f007:**
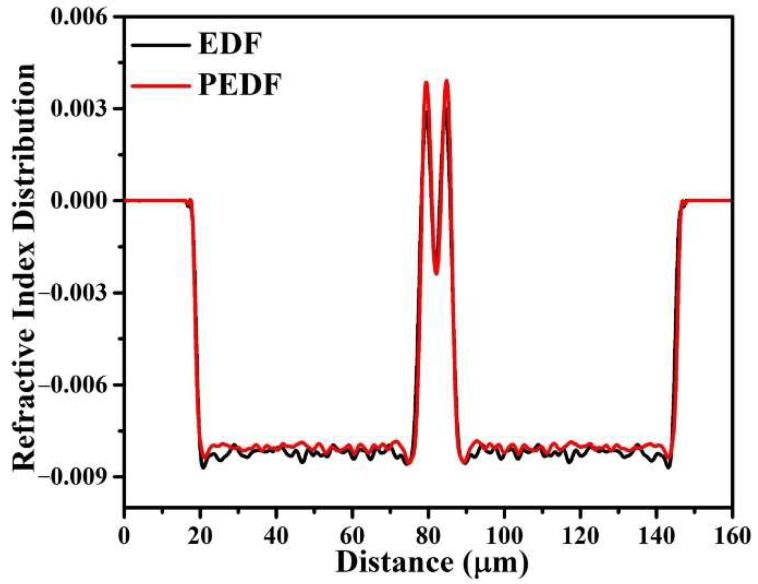
The refractive index difference distribution of EDF and PEDF.

**Figure 8 materials-15-06090-f008:**
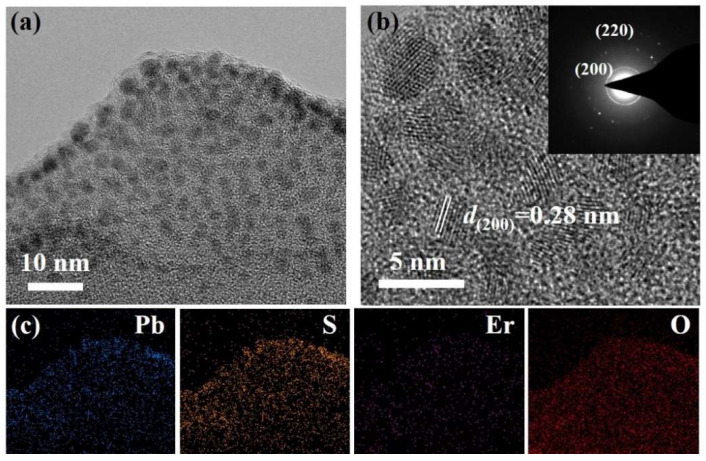
(**a**) TEM image of PEDF; (**b**) HRTEM image of PEDF, and the inset is the corresponding SAED; (**c**) elemental mapping of PEDF.

**Table 1 materials-15-06090-t001:** Energy parameters of different doping sites in the 3MR microstructural models.

Microstructures	Model (a)	Model (b)	Model (c)
Energy (a.u.)	7.3141	6.0843	7.3265

**Table 2 materials-15-06090-t002:** Excited states parameters of PbS/Er-3MR local structural model (a).

Electronic Transition	Excitation Energies (eV)	Oscillator Strength (f)	Wavelength (nm)
S_0_–S_1_	0.7878	0.0002	1573.83
S_0_–S_2_	1.2934	0.0009	958.62
S_0_–S_3_	1.3664	0.0005	907.39
S_0_–S_4_	1.5441	0.0017	802.93
S_0_–S_5_	1.8694	0.0009	663.24

**Table 3 materials-15-06090-t003:** Excited states parameters of PbS/Er-3MR local structural model (b).

Electronic Transition	Excitation Energies (eV)	Oscillator Strength (f)	Wavelength (nm)
S_0_–S_1_	0.7770	0.0043	1595.77
S_0_–S_2_	0.8086	0.0043	1533.41
S_0_–S_3_	1.2842	0.0008	965.48
S_0_–S_4_	1.3535	0.0022	916.05
S_0_–S_5_	1.5661	0.0031	791.67
S_0_–S_6_	1.9044	0.0016	651.03

**Table 4 materials-15-06090-t004:** The content of different elements in the fiber samples.

Samples	O (wt.%)	Ge (wt.%)	Er (wt.%)	Al (wt.%)	Si (wt.%)	Pb (wt.%)
EDF	42.96	6.85	0.12	0.61	49.46	0
PEDF	42.69	6.99	0.12	0.62	48.59	0.98

## Data Availability

Not applicable.

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
