# Peer review of "Improved Fluorescence and Gain Characteristics of Er-Doped Optical Fiber with PbS Nanomaterials Co-Doping"

_materials, 2022, doi:10.3390/ma15176090_

Round 1

Reviewer 1 Report

I attached referee report in the separated sheet.

Reviewer 2 Report

The paper is related to improvement of Er-doped optical fibers, which are very important for optical communication. This is an important problem and the research is well motivated. The first part is theoretical and it is based on DFT analysis. This part looks strong and informative and no extra revisions are required. The second of the paper is experimental and here some improvements can be recommended.

1. It is recommended briefly explain what is MCVD. Although the references are available, some explanation in the next would be useful.

2. Fig. 5 Please change "Refraction index" to "Refractive index".

3. What is the porous layer deposited on inner" wall ? 

4. Schematic drawing of silica tube after deposition and annealing would be useful.

5. The silica tube has external diameter 30 mm and internal diameter 25 mm. Please show on schematic drawing "core and cladding of PEDF, which are 8.85 and 125.28 um thick".
